# Identifying Early-Life Behavior to Predict Mothering Ability in Swine Utilizing NU*track* System

**DOI:** 10.3390/ani13182897

**Published:** 2023-09-13

**Authors:** Savannah Millburn, Ty Schmidt, Gary A. Rohrer, Benny Mote

**Affiliations:** 1Department of Animal Science, University of Nebraska-Lincoln, Lincoln, NE 68588, USA; 2United States Meat Animal Research Center, United States Department of Agriculture-Agricultural Research Service, Clay Center, NE 68933, USA

**Keywords:** tracking individual pigs, automation of behavioral monitoring, livestock precision farming

## Abstract

**Simple Summary:**

Improving sow productivity and welfare have been long-withstanding goals for the swine industry. Identifying novel traits and phenotypes to accomplish these objectives is needed. The objective of the current study was to determine if activity-based phenotypes collected by the NU*track* livestock monitoring system, NU*track*, could serve as early-life indicator traits for mothering ability in swine. The phenotypes collected included distance traveled, average velocity, angle rotated, time allocated to eating, lying lateral, lying sternal, standing, and sitting. The response variables selected in first parity females to model mothering ability were gestation length, number born alive, and number weaned. Simple linear regression models were generated to analyze the relationship between activity traits and reproductive measures. The results of this study indicate that select activity traits may be used to explain a portion of the variability in gilt reproductive performance. This information is foundational to informing future selection decisions pertaining to the use of activity traits in breeding programs.

**Abstract:**

Early indicator traits for swine reproduction and longevity support economical selection decision-making. Activity is a key variable impacting a sow’s herd life and productivity. Early-life activities could contribute to farrowing traits including gestation length (GL), number born alive (NBA), and number weaned (NW). Beginning at 20 weeks of age, 480 gilts were video recorded for 7 consecutive days and processed using the NU*track* system. Activity traits included angle rotated (radians), average speed (m/s), distance traveled (m), time spent eating (s), lying lateral (s), lying sternal (s), standing (s), and sitting (s). Final daily activity values were averaged across the period under cameras. Parity one data were collected for all gilts considered. Data were analyzed using linear regression models (R version 4.0.2). GL was significantly impacted by angle rotated (*p* = 0.03), average speed (*p* = 0.07), distance traveled (*p* = 0.05), time spent lying lateral (*p* = 0.003), and lying sternal (0.02). NBA was significantly impacted by time spent lying lateral (*p* = 0.01), lying sternal (*p* = 0.07), and time spent sitting (*p* = 0.08). NW was significantly impacted by time spent eating (*p* = 0.09), time spent lying lateral (*p* = 0.04), and time spent sitting (*p* = 0.007). This analysis suggests early-life gilt activities are associated with sow productivity traits of importance. Further examination of the link between behaviors compiled utilizing NU*track* and reproductive traits is necessitated to further isolate behavioral differences for potential use in selection decisions.

## 1. Introduction

Selection of replacement females is a critical decision for commercial producers, sometimes based on little-known information outside of pedigrees and physical conformation [1]. From an economic standpoint, maintaining a productive, healthy sow herd while placing emphasis on female longevity is of high importance [2,3,4]. However, sow longevity and productivity can be ambiguously defined terms across different production systems. Further, these traits are not observed until advanced stages of life and have low heritability measures increasing investment costs in replacement gilts [5,6,7,8]. Breeding failures, lameness, and poor performance in the farrowing crate have historically been implicated as reasons for early culling [9]. Determining a connection between early-life observable activities and reproductive traits could give commercial producers the ability to improve sow herds in a more rapid and cost-effective manner.

Gestation length (GL) measures the amount of time between when a sow is bred and when she farrows. Shorter gestation lengths have been suggested to be significantly associated with increased log odds of stillbirths in addition to decreased piglet viability [10]. Conversely, increased GL is positively correlated with piglet birth weight but negatively correlated with pigs farrowed per litter as well as total litter birth weight [11]. Moreover, a genome-wide association study assessing gestation length across a range of parities discovered differences between QTL’s shared for different parities, adding to the complexities of this trait [12]. Thus, breeding goals related to GL must be defined within the framework of production goals specific to an operation. Similarly, past research on number born alive (NBA) provides mixed results, with certain studies suggesting gilts raised in larger litters endure lasting negative environmental effects while other studies show that females from small litters of origin were culled earliest and produced small litters [13,14,15]. Moreover, preweaning mortality may result from negative maternal behaviors such as impartiality to piglet distress calls, method of laying down, and savaging, negatively impacting total number weaned (NW) [16,17,18]. Additionally, poor NW can be the outcome of management failures or environmental conditions unrelated to the sow herself. In addition to the complex associations above, NBA and NW are lowly heritable traits, with reported average heritabilities of 0.07 and 0.06, respectively [19]. Gestation length has been reported to be moderately heritable, at 0.29 when estimated for a Landrace population and 0.34 in a Yorkshire population [20].

Previous research in swine has failed to quantify activity and behavioral traits consistently and objectively. Methods used to categorize animal behavior and response type include back and resident-intruder tests, scale activity scores, and post-farrowing postural scoring. Both back test outcomes and scale activity scores have been described to be heritable traits [21,22]. However, inconsistencies in repeated back tests call into question the stability of behavioral differences across time [23]. Sow postural scoring studies have reported litter size differences between sows categorized as “crushers” vs. “non-crushers” [24]. However, underlying physiological or genetic differences between these sows have not been examined. Thus, behavioral studies have been limited by subjectivity, sample size, and the time and labor involved in data collection.

As behavior studies have been limited in size and scope, few data exist examining the impact of general activity traits during the gilt growing phase on later observed reproductive traits such as GL, NBA, and NW that are key reproductive traits in sow longevity. The objective of this study was to determine if there is a relationship between activity traits expressed early in life and reproductive traits measured during the first farrowing event. Further, the goal was to verify the NU*track* system’s ability to objectively measure gilt activity traits in a group-housed environment. We hypothesized that active and passive activities displayed during the gilt growing phase may be indicators of parity one performance.

## 2. Materials and Methods

### 2.1. Animals

All procedures involving the use of animals were approved by the University of Nebraska Institutional Animal Care and Use Committee protocol number 2089. The group-housed replacement gilts (*n* = 2859) used in this dataset were housed at the United States Meat Animal Research Center (USMARC) in Clay Center, NE. Replacement gilts were all produced on site and with breed compositions of Yorkshire by Landrace. The USMARC swine resource population is managed as a rotational crossbreeding herd alternating use between Yorkshire and Landrace semen. All semen is sourced from four separate commercial genetics suppliers. Replacement gilts are kept in groups of 12–16 gilts in finishing barn pens that are 2.4 × 7.0 m. Gilts were observed and kept for use as replacements or culled from the herd by an experienced caretaker on the primary basis of conformation at approximately 165 days of age. Replacement gilts are managed in facilities reflective of commercial production with newly constructed breed, group-housed gestation housing approximately 45 females per pen with 1 electronic sow feeder, and farrowing barns equipped with 20 farrowing crates housing 1 week of farrowings.

### 2.2. Data Collection

Video recording of gilts began at approximately 20 weeks of age. Gilts were under cameras for nine consecutive days. Only full 24 h cycles were considered for analysis, resulting in 2859 individual gilts with observation lengths ranging from 6 to 8 full days under cameras. Any gilts that were removed due to morbidity or mortality during the trial period were consequently removed from analysis. The age of gilts at observation was age-matched to the timing of replacement and culling decisions in typical commercial swine operations. Lorex (Linthicum, MD, USA) cameras at USMARC were procured and installed by University of Nebraska-Lincoln researchers. Video output was managed and analyzed at UNL.

The NU*track* system was used for continuous observation of gilts in this study. NU*track* is a deep-learning-based multi-object tracking system that can achieve >95% precision and recall tracking the long-term location and identity of individual pigs in group-housed settings [25]. An FLIR/Lorex 4k Ultra HD NVR System with infrared capability was used to record video at 5 frames per second. These cameras are cost-effective and capable of withstanding a wide range of environmental conditions. Video captured by the cameras was downloaded to a Dell-Alienware GPU (Round Rock, TX, USA) with an NVIDIA Graphics System (Santa Clara, CA, USA) for processing and analysis. Cameras were positioned downward facing from the ceiling in the center of a pen with care to avoid feed lines and piping that may downgrade images of gilts in pens. In this study, one camera was used to capture continuous video recordings of animals in each pen. For a thorough technical description of the NU*track* system, reference Psota et al. (2019) [25]. In previous research, NU*track* was used to analyze nursery and finishing pigs [25]. The size of gilt analyzed in this study mirrors the approximate size of finishing pig that NU*track* was previously trained on. In brief, this method for long-term tracking of individual animals within a group-housed setting is feasible via deep convolutional neural networks. These networks locate individual targets (animals) within a fixed living space and classify their identity [25]. This detection utilizes deep learning from human annotated images, in which it joins body parts (left ear, right ear, shoulder, and tail) into “instances” via part association vectors [25]. Additionally, activity tracking is processed via the Bayesian multi-object tracking method. Utilizing frame-to-frame movement probabilities with images being captured at 5 frames per second, probabilities of individual identification (by the NU*track* ear tags) can be assessed. Across a variety of environments, this method has shown to be over 90% accurate in maintaining individual identification [25]. In short, NU*track* does not classify pig behavior, instead, it tracks activity and body posture determined by body part position and change in location based on frame-to-frame movement analysis to verify the identities of individuals in each pen, and 16 unique Hog Max tags from Allflex (Rahway, NJ, USA) were assigned per pen for this study. Tags used were non-barcoded and non-radio frequency identification. Ear tag color and alphanumeric sequences were generated for this study to maximize tag identification probability.

NU*track* has capabilities to continuously track a range of social and normative behaviors and activities [26]. For the purposes of this study, the recorded activities include angle rotated (radians), average velocity (meters/second), distance travelled (meters), time spent eating (seconds), time spent lying lateral (seconds), time spent lying sternal (seconds), time spent sitting (seconds), and time spent standing (seconds), which were all trained on human annotated images and videos and validated such that they can be considered to be as close to ground truth as possible using an automated algorithm. Angle rotated represents the radians of rotation an individual pig makes per day. In addition to individual activities, location within pen and proximity to pen mates (meters) were recorded. In this study, distance traveled, average velocity, body angle rotated, time spent eating, and time spent standing were considered “active” traits. Time spent lying lateral, time spent lying sternal, and time spent sitting were categorized as “passive traits”.

In addition to NU*track* data, outdoor ambient temperature was obtained for analysis. Temperatures were acquired from the Department of Natural Resources at the University of Nebraska-Lincoln through the Nebraska Mesonet network. The weather station where data were collected is located on site at USMARC. Air temperatures were automatically taken once per minute and averaged to generate a 24 h average temperature. Average temperatures for each day on test as well as 3 days prior to test were included in the analysis. The outside air temperature was utilized for analysis since finishing units at USMARC are not equipped with cooling cells and are maintained at minimum temperatures during cold weather. The set barn temperature at USMARC is approximately 18 °C. All outside air temperatures below 18 °C in the dataset were set equal to the barn set point for analysis. The dataset includes one spring, two summers, two falls, and two winters.

Farrowing records were collected at USMARC on a subset of gilts that were retained following observation under NU*track* cameras. Normal farrowing assistance, such as sleeving a sow if it had been greater than 30 min since last live birth, was provided in accordance with USMARC standard procedures during working hours (06:00–15:00 h). Day of birth for piglets born overnight was estimated based on dryness of umbilical cords and membranes as well as activity of piglets. Sows were required to stand at least two times each day to check for health and wellness by caretakers. In total, 480 gilts were considered after censoring of farrowing records. Two gilts were removed from the dataset for failure to wean any live pigs during the first farrowing event. Traits reported for each parity included gestation length (days), number born alive, litter birth weight (kg), mean birth weight (kg), number weaned, litter weaning weight (kg), and mean litter weaning age. Gestation was computed as days between first insemination and farrowing. Parturition was induced at 116 or 117 days of gestation on 103 of the 480 animals (21.4%). Cross-fostering was conducted when females had more than 12 live piglets at 1 day of age. Extra piglets removed were selected to match the weight of piglets in the pens the animals were fostered into. Only male piglets were fostered, and they were placed in litters with the fewest piglets of comparable age available. Creep feed was provided beginning at 10 days of age. Average age at weaning was 26.7 days. Number weaned is the number of piglets birthed by the pig that survived to weaning.

### 2.3. Statistical Analysis

Data were analyzed using linear regression models (R version 4.0.2). Non-linear trends were not seen in the data, and reproductive and activity variables were left untransformed for analysis. All activity traits determined by NU*track* were averaged across days to represent a single 24 h period for each individual gilt. Linear regressions were initially fit for all seven reproductive traits in the dataset. Gilt weight, ambient temperature, loin muscle area, and 10th rib backfat thickness were initially included in the analysis. However, these traits were ultimately dropped from the final regression models as they were not significant (*p* > 0.10). Simple linear regression models were favored due to the multicollinearity between NU*track* traits*,* ease of interpretation of the association between NU*track* traits and farrowing performance, and the units of measurement for the activity traits. Variable selection was performed based upon linear regression models generated using the lm() function in R that were significant. Reproductive traits selected for in-depth analysis included gestation length (GL), number born alive (NBA), and number weaned (NW) using the model:Y = Xb + e
where Y is one of the reproductive traits studied (GL, NBA, or NW), X is one of the eight NU*track* measurements, b is the simple linear regression coefficient associating the NU*track* measurement with the reproductive trait, and e is a random error. For linear regression, *p*-values ≤ 0.10 were considered significant. Correlations among NU*track* measurements were calculated using the cor() function. Model comparison was performed using the anova() function.

## 3. Results

Summary statistics were calculated for each farrowing trait considered in the analysis as well as the activity traits. Parity one gestation length had a mean of 115.2 days, standard deviation of 1.5, and median value of 115 days. Number born alive resulted in a mean of 11.7 piglets per sow, standard deviation of 3.3, and median value of 12 piglets. The average number of piglets weaned in parity 1 was 10.3 piglets, with a standard deviation of 3.2 and a median of 11 piglets. Complete summary statistics for reproductive traits are reported in Table 1, and complete summary statistics for activity traits are reported in Table 2. Correlations between all activity traits are calculated and reported in Table 3.

Twenty-four simple linear regression models were estimated individually regressing all eight activity traits (angle, average speed, distance traveled, eat, lie lateral, lie sternal, sit, stand) on gestation length, number born alive, and number weaned. All linear regression estimates and respective statistical significance can be found in Table 4. Gestation length, angle rotated, average speed, distance travelled, time spent lying lateral, and time spent lying sternal were found to be statistically significant. Time spent lying lateral, time spent lying sternal, and time spent sitting were statistically significant with respect to number born alive. Time spent eating, time spent lying lateral, and time spent sitting were statistically significant in regression models for number of piglets weaned.

Of the eight separate models fitted for gestation length, the following five traits were found to be statistically significant: angle rotated, average speed, distance travelled, and time spent laying lateral and sternal. It was determined that angle rotated was a marginally significant predictor of gestation length (r^2^ = 0.009, β_1_ = 0.000396, *p* = 0.038). Average speed travelled was a marginally significant predictor of gestation length (r^2^ = 0.007, β_2_ = 9.988, *p* = 0.065). Distance travelled was marginally significant in predicting gestation length (r^2^ = 0.008, β_3_ = 0.000563, *p* = 0.051). Time spent lying lateral significantly predicted gestation length (r^2^ = 0.019, β_5_ = −0.0000449, *p* = 0.002). Finally, time spent lying sternal significantly predicted gestation length (r^2^ = 0.012, β_6_ = 0.00003723, *p* = 0.016).

Simple linear regression showed that time spent lying lateral, lying sternal, and sitting were significant predictors of the number born alive in parity one. Time spent lying lateral significantly predicted the number born alive (r^2^ = 0.014, β_5_ = 0.0000858, *p* = 0.01). Time spent lying sternal was a marginally significant predictor of the number born alive (r^2^ = 0.007, β_6_ = −0.0000613, *p* = 0.07). Time spent sitting was a marginally significant predictor of the number born alive (r^2^ = 0.006, β_7_ = −0.000368, *p* = 0.08).

Finally, simple linear regression models were generated for parity one number weaned. Activity traits that were significant predictors of the number weaned included time spent eating, lying lateral, and sitting. Time spent at the feeder was a marginally significant predictor of number weaned (r^2^ = 0.006, β_4_ = −0.000175, *p* = 0.09). Time spent lying lateral was a marginally significant predictor of number weaned (r^2^ = 0.009, β_5_ = 0.0000684, *p* = 0.03). Finally, time spent sitting significantly predicted number weaned (r^2^ = 0.015, β_7_ = −0.000557, *p* = 0.006).

Independent variables for multiple linear regression models were selected based on the results of simple linear regression. When two highly correlated NU*track* traits (absolute value of correlation > 0.80; Table 3) were significant for a reproductive trait, only the most significant activity trait was included. For gestation length, the selected multiple regression model included time spent lying lateral and angle rotated. The full model was statistically significant (r^2^ = 0.0046, β_lateral_ = −0.0000395, β_angle_ = 0.000267, *p* = 0.0046), however, time spent lying lateral was the sole statistically significant predictor variable (*p* = 0.0109). The results of the model comparison showed that the multiple regression model was not a significant improvement over a simple regression model for time spent lying lateral (*p* = 0.175).

The multiple regression model selected for number born alive included time spent lying lateral and time spent sitting. The complex model was statistically significant (r^2^ = 0.014, β_lateral_ = 0.0000802, β_sit_ = −0.00031, *p* = 0.0129), but time spent lying lateral was the only significant independent variable (*p* = 0.017). The model comparison indicates that multiple regression was not a significant improvement over a simple model using time spent lying lateral (*p* = 0.1446).

The multiple regression model selected for number weaned included time spent lying lateral, time spent sitting, and time spent eating. The complex model was statistically significant (r^2^ = 0.0186, β_lateral_ = 0.00004792, β_sit_ = −0.0005178, β_eat_ = −0.0001266, *p* = 0.00755), but time spent sitting was the only significant independent variable (*p* = 0.0123). Model comparison indicated that multiple regression was not a significant improvement over a simple model using time spent sitting (*p* = 0.0968).

## 4. Discussion

The results of this analysis indicate potential for early-life activity traits to serve as additional indicators of parity one reproductive performance beyond what has been known to the scientific community to date. Moreover, activity trait distribution and statistical significance validated the consistency of NU*track* as a continuous monitoring system for group-housed livestock. For parity one gestation length, three active traits (angle rotated, average speed, distance traveled) were individually reported to be significant predictors of length of gestation. Further, two passive traits (time spent lying lateral and lying sternal) were significant predictors of gestation length. Three passive NU*track* traits (time spent lying lateral, lying sternal, sitting) were significant predictors of parity one number born alive. Additionally, one active trait (time spent eating) and two passive traits (time spent lying lateral, sitting) were significant predictors of parity one number weaned. These outcomes suggest that there is a link between early-life behavior and parity one reproductive traits.

Gestation length was the only parity one reproductive trait shown to be significantly predicted by more than one active NU*track* trait (body angle rotated, average velocity, distance traveled). These three NU*track* traits were individually shown to have a positive impact on parity one gestation length. To the authors’ knowledge, no prior research has analyzed the link between early-life “active” traits and parity one farrowing records. The results of the simple regression models suggest that more active gilts tend to have longer parity one gestation lengths. Moderate heritabilities (*h*^2^ = 0.29) for gestation length have been reported in the literature [20]. Similarly, heritabilities calculated from NU*track* activity data for angle rotated, average speed, and distance traveled have been reported as 0.31, 0.32, and 0.32, respectively [27]. The moderate heritabilities reported for activity traits imply there is potential for gestation length to be altered by selection on NU*track* activity traits.

In addition, gestation length, number born alive, and number weaned were shown to independently be significantly predicted by multiple “passive” NU*track* traits. Time spent lying lateral had a negative impact on length of gestation while time spent lying sternal had a positive impact on gestation length. It is important to note that based on these results, a model combining lying lateral and lying sternal into one lie trait would be counter-productive due to the divergent signs on the regression coefficients and their strong negative phenotypic correlation of −0.80. Time spent lying lateral (+), lying sternal (−), and sitting (−) were shown to have a significant impact on the prediction of the number born alive. Further, time spent eating (−), lying lateral (+), and sitting (−) were modeled as having a significant impact on the prediction of number weaned.

The multiple regression models that were generated for GL, NBA, and NW indicate that individual key variables accounted for a majority of the variation in these reproductive traits. A simple model for time spent lying lateral was the best predictor of GL. This indicates that of the tested NU*track* variables, time spent lying lateral, should be viewed as the most valuable predictor variable for gilt gestation length. Similarly, for NBA, a simple model using time spent lying lateral was shown to be the best fit. The number weaned was the only variable that was best predicted by an independent variable besides time spent lying lateral. For NW, time spent sitting was the best predictor. However, the limited variation in the distribution of time spent sitting necessitates interpreting the results from this dataset with caution.

Across the simple regression models, time spent lying lateral has a consistent, statistically significant impact on GL, NBA, and NW. The impact that lying lateral exerts on NBA and NW is consistent and positive, indicating that gilts who spend a greater amount of time lying laterally during the growing phase farrowed and weaned a greater number of pigs in their first farrowing event. When viewing decreased GL favorably given the negative correlations between GL and piglets farrowed per litter, time spent lying lateral is favorably related to parity one production. The consistency of time spent lying lateral across GL, NBA, and NW indicates that gilts who spend a greater amount of time lying lateral at an early age are more productive mothers with more favorable outcomes in their first parity.

Conversely, increased time spent lying sternal was unfavorably associated with reproductive traits. Further, time spent lying sternal and time spent lying lateral were consistently antagonistic in their effect on the three reproductive traits examined. The correlation between lying lateral and lying sternal was strong and negative (r = −0.80). Past NU*track* analysis has viewed total time lying as a function of combined time spent lying lateral and time spent lying sternal. The consistent opposite effect of these traits paired with the strong negative correlation suggests lying lateral and lying sternal should be viewed as separate traits and not components of total time lying. The lower calculated heritability for total lying time (0.21) compared to other NU*track* traits adds support to the theory that lateral lying and sternal lying should be viewed as separate activity traits [27]. Determining potential underlying mechanisms that explain differences in time allocation and preference of lying lateral versus lying sternal should be explored.

Sow postural changes in the farrowing crate have been used to categorize risk to piglets. Although not a direct parallel, it has been shown that sows exhibiting restless behavior following farrowing were more often categorized as savagers and aggressive towards their litters [16,28]. If gilts who spend a greater amount of time lying lateral also exhibit fewer postural changes in the crate, time spent lying lateral could serve as an indicator trait for post-farrowing behavior. Similarly, if time spent lying sternal is reported to be an indicator of more alert or restless “personalities”, this trait could be used to select for females that will be more docile during their lifetime. Future research should examine farrowing behavior and postural changes in gilts who spend a significant amount of time lying lateral or lying sternal. Determining a potential link between postural changes (lying lateral, lying sternal) during the growing phase and the propensity to frequently change postures or exhibit aggressive behavior in the farrowing crate could demonstrate a stability in behavior or personality type in swine across life stages. A better understanding of the genetic and behavioral components associated with savaging and agonistic mothering behavior would result.

Reported regression coefficients for the activity traits listed in Table 4 were relatively low. Noting unit of measurement for each trait can assist in more practical interpretation of regression results. Angle rotated was measured in radians and average speed was represented as meters per second. Distance travelled was reported as the number of meters travelled, on average, per day of observation. Given the diminutive size of units and nature of 24 h observation, scaling of regression coefficients for interpretation is relevant. Interpretation of the lateral–sternal complex is pertinent when rescaling coefficients to an hourly basis. A 1 h increase in daily time spent lying lateral during the growing phase resulted in a 3.9 h decrease in gestation length. Conversely, a 1 h increase in time spent lying sternal resulted in a 3.2 h increase in gestation length. Similar results can be seen for both NBA and NW. A 1 h daily increase in time spent lying lateral during the growing phase resulted in 0.31 more piglets born alive in the first farrowing event. Again, the same increase in time spent lying sternal resulted in an average decrease of 0.21 piglets born alive. Finally, a 1 h increase in time spent lying lateral resulted in an average increase of 0.25 piglets weaned. A 1 h increase in time spent lying sternal resulted in 0.11 fewer piglets weaned. Across all three reproductive variables included in the analysis, lying lateral and lying sternal have a similar magnitude but antagonistic effects on outcomes. Recognizing the relationship between these two activity traits is the most striking outcome of this analysis. These results indicate that passive NU*track* lying traits offer insight into differences in reproductive traits.

An important limitation to note is that gestation length was measured on a discrete basis during the hours that farm staff was present in facilities. The discrete nature of this variable is a limitation when examining the impact of NU*track* activity traits (measured in seconds) on parity one gestation length. While the gestation length observations are relatively precise by day, continuous hourly observation of farrowing events would yield a more accurate dependent variable to model. A core assumption of this study was the accuracy of activity trait measurement using the NU*track* system. Though NU*track*’s activity tracking was trained on human annotated images and videos with human validation of the training data, some ethologists might not consider the data generated for this manuscript as ground truth given it is impractical to human validate each second of this dataset, and the authors do not see this as a limitation of these models as several significant associations were detected.

Research has reported that highly productive sow herds record high number born alive and low preweaning mortality rates [29]. Further studies have reported a negative correlation between gestation length and NBA or NW but positive correlations between GL and piglet birth weight [11,30]. Clearly, the genetic and environmental elements contributing to reproductive traits are intricate, with the results presented herein adding additional information and considerations for reproductive success. It follows that improvement in farrowing traits will likely be the result of incremental changes to breeding programs and selection decisions. The simple linear regressions generated in this analysis resulted in low r^2^ values, ranging from 0.006 to 0.019 across significant models. Although the explanatory power of NU*track* traits in these models was relatively low, the traits considered must be contextualized within the larger framework of swine breeding systems. In theory, explaining 1% of a lowly heritable reproductive trait with moderately heritable, novel activity traits observed early in life (≤150 days) offers potential to make earlier selection decisions, thus generating more robust breeding programs. Inclusion of multiple activity traits in a selection index where each trait is appropriately weighted to account for the genetic and phenotypic correlations should increase the rate of genetic change observed for first litter reproductive performance. However, in future research, it is important to quantify a genetic link between activity traits and farrowing traits.

## 5. Conclusions

The results of this analysis showed a connection between the objectively measured NU*track* activity traits collected during the growing phase and parity one gestation length, number born alive, and number weaned. Further, these results validate the NU*track* system’s ability to detect and continuously monitor group-housed livestock activity. The regression models in this study show a significant linear relationship between the activity traits and parity one farrowing records, suggesting potential for early-life activity to serve as an indicator for parity one farrowing performance. In addition, multiple regression models were tested for each reproductive trait. The results of model comparison indicate that each reproductive trait is best predicted by a single NU*track* trait. Overall, increased time spent lying lateral had a consistent, favorable effect on reproductive outcomes. In addition, increased time spent lying sternal had a stable, negative impact on reproductive outcomes. These results suggest that time spent lying lateral and time spent lying sternal could serve as important early-life behavioral indicators of mothering ability. Further studies utilizing increased sample size and hourly gestation length records should be conducted to improve accuracy of gestation length models. Moreover, additional research is needed to determine whether the relationships established in this study are genetic or environmental in nature.

## Figures and Tables

**Table 1 animals-13-02897-t001:** Summary statistics for parity one gilt (*n* = 480) reproductive traits.

Trait	Mean	Standard Deviation	Minimum	Maximum
Gestation Length	115.2	1.50	111	119
Number Born Alive	11.7	3.33	1	19
Number Weaned	10.34	3.24	1	18
Herd Life	2	0.85	1	3

**Table 2 animals-13-02897-t002:** Summary statistics for parity one gilt (*n* = 480) activity traits. All values for traits measured in seconds are rounded to the nearest second.

Trait	Mean	Standard Deviation	Minimum	Maximum
Angle (rad)	1446.7	357.4	489.3	2639.5
Speed (m/s)	0.074	0.0126	0.041	0.137
Distance (m)	947.2	237.4	335.1	1923.9
Eat (s)	5827	1415	1886	12,229
Lie Lateral (s)	40,392	4546	26,848	53,103
Lie Sternal (s)	29,866	4441	15,719	45,403
Sit (s)	1016	717	102	5308
Stand (s)	15,126	2825	6186	22,891

**Table 3 animals-13-02897-t003:** Correlations (*n* = 480) between NU*track* activity traits.

Trait	Angle	Speed	Dist.	Eat	Lat.	Stern.	Lie Total	Sit	Stand
Angle	1								
Speed	0.59	1							
Distance	0.97	0.65	1						
Eat	0.28	0.12	0.29	1					
Lie Lateral (s)	−0.26	−0.005	−0.24	−0.29	1				
Lie Sternal (s)	−0.22	−0.05	−0.22	−0.08	−0.80	1			
Lie Total	−0.75	−0.08	−0.72	−0.58	0.35	0.29	1		
Sit	0.16	0.10	0.10	0.02	−0.11	−0.002	−0.18	1	
Stand	0.72	0.05	0.71	0.58	−0.33	−0.29	−0.97	−0.07	1

**Table 4 animals-13-02897-t004:** Regression estimates of reproductive traits with angle rotated, average speed traveled, distance traveled, time spent eating, laying lateral, laying sternal, and standing.

Parity 1 Regression Estimates			
Activity Trait	Gestation Length	Number Born Alive	Number Weaned
Angle (rad)	0.000396 **	−0.000546	−0.000496
Speed (m/s)	9.988 *	2.823	12.528
Distance (m)	0.000563 **	−0.000292	−0.000195
Eat (s)	−0.00000581	−0.000169	−0.000175 *
Lie Lateral (s)	−0.0000449 ***	0.0000858 ***	0.0000685 **
Lie Sternal (s)	0.0000372 ***	−0.0000613 *	−0.0000308
Sit (s)	−0.0000258	−0.000368 *	−0.000557 ***
Stand (s)	0.0000261	−0.000047	0.0000655

* *p* ≤ 0.10; ** *p* ≤ 0.05; *** *p* ≤ 0.01.

## Data Availability

Data available upon request.

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
