# Peer review of "Identifying Early-Life Behavior to Predict Mothering Ability in Swine Utilizing NUtrack System"

_animals, 2023, doi:10.3390/ani13182897_

Round 1
Reviewer 1 Report
Review
Identifying Early-Life Behavior to Predict Mothering Ability in 2 Swine Utilizing NUtrack System
Comments:
The objective of the present study was to determine if there is a relationship between activity traits expressed early in life and reproductive traits measured during the first farrowing event.
It is an interesting manuscript, well-written and properly organized. I just made a few comments below.
1- In the Introduction section, it is pretty clear that the sow replacement is not easy since the number of variables that interfere with the process is high. It is not clear in the methods the link between the activity of the animals and the variables that predict the sow selection for a good performance. I believe the authors should elaborate more on this matter.
2- A research schematic must be added here to help the reader understand the whole process without checking the previously published paper on the concept ( lines 123-125).
3- The authors agree with comment 1 I made since in lines 354-355 they state "Clearly, the genetic and environmental elements contributing to reproductive traits are intricate." Therefore, the traits associated with the activities recorded and analyzed should be clearer.
My recommendation is a minor revision.
Author Response
- In the Introduction section, it is pretty clear that the sow replacement is not easy since the number of variables that interfere with the process is high. It is not clear in the methods the link between the activity of the animals and the variables that predict the sow selection for a good performance. I believe the authors should elaborate more on this matter.
Added clarity to introduction pointing to limited prior behavior studies and reproductive traits included in sow longevity.
- A research schematic must be added here to help the reader understand the whole process without checking the previously published paper on the concept ( lines 123-125).
- Reworded section to give a high level understanding of the process NUtrack utilized to identify individual animals and track posture.
- The authors agree with comment 1 I made since in lines 354-355 they state "Clearly, the genetic and environmental elements contributing to reproductive traits are intricate." Therefore, the traits associated with the activities recorded and analyzed should be clearer.
- We certainly agree that many traits (genetic and environmental) impact reproductive success. We opted to not include the vast options of known contributions to reproductive success such that we could draw attention to the activity traits that have never been studied to this detail and scope before. Added a statement to the sentence to note we are attempting to add to what we already know affects reproductive success.
Reviewer 2 Report
The Materials and Methods require more information on what constitutes USMARC standard protocols for farrowing assistance. Important variables include: a) were any of sows induced? b) did sows undergo treatment for low feed intake events c) were litters cross fostered d) were sows required to get up once a day and if so, was it standardized. Mortality was 12% - was there a correlation between lying time and reason for pigs weaned?
Results: If gestation length is determined by human observation during 6 am to 15:00 hours, then statistical inference based on gestation length of a few hours can be incorrect. Do you have substantiated information on the gestation length or include only animals whereby gestation length was observed and recorded?
Psota's papers reference tracked nursery and finishing pigs not gravid or lactating females. Did human annotation provide ground truth of positions and posture of sows for training dataset?
Author Response
The Materials and Methods require more information on what constitutes USMARC standard protocols for farrowing assistance. Important variables include: a) were any of sows induced? b) did sows undergo treatment for low feed intake events c) were litters cross fostered d) were sows required to get up once a day and if so, was it standardized. Mortality was 12% - was there a correlation between lying time and reason for pigs weaned?
a. Standard farrowing assistance was clarified. Induction numbers were given. Notation of crossfostering procedures was outlined. Standardized twice daily standing sow health checks were noted. No sows were treated for low feed intake. Assuming the reviewer is referring to 12% preweaning mortality, the sample size by mortality reason was unfortunately too small to analyze in this data set. The PRRS break that the farm experienced excluded the opportunity to include additional parities into this analysis.
Results: If gestation length is determined by human observation during 6 am to 15:00 hours, then statistical inference based on gestation length of a few hours can be incorrect. Do you have substantiated information on the gestation length or include only animals whereby gestation length was observed and recorded?
a. Included additional information on how farrowing data was justified when litters were born outside of staffed hours. Given the significant association reported in this manuscript, future work to more accurately access exact start of parturition should include the use of 24 hour human or video surveillance.
Psota's papers reference tracked nursery and finishing pigs not gravid or lactating females. Did human annotation provide ground truth of positions and posture of sows for training dataset?
a. The gilts in this study were analyzed at the same weight/maturity as the finishing pigs the system was trained on which did use human annotation to provide ground truth. If/when the system is used to analyze mature animals, the system will be revalidated to ensure the system continues to maintain identity and accurately call posture on mature animals.
Reviewer 3 Report
The study presented by Millburn et al. aims at examining the association between early-stage activities of gilts and reproductive traits at later life stages. This work is of great interest to swine farming, management, and breeding. The introduction section is appropriate, logical, and sound. The objective of the study is well-defined. However, in my perspective, the statistical analysis and/or regression model setup can be improved, while the results might be updated depending on the revised analysis. My specific comments are below.
Lines 132-135: the measures and quantifications are collected by deep learning models. I would suggest the authors indicate that the measures are not exactly the same as ground truth measures.
Line 161: presenting the linear equations in a formula form is encouraged. For the linear model, did it account for effects in addition to the activity variables e.g., social group and body weight (or weight gain) during the observation period at the early stage? If yes, please list all the effects included in the model. If not, please justify the reason for not adjusting for those variables. Overall, there is a lack of information to justify the statistical model.
Line 187-189: For the highly correlated variables e.g., angle and distance (referred to correlations shown in Table 3), what is the reason for including both variables in the regression model? Please justify.
Line 244-245: the results based on the statistical analysis described in this manuscript may be inaccurate. There was a time difference between the measures of the early-stage activities and the measures of the reproductive traits. It would be risky to omit effects such as animal ID, social group, weight (or weight gain), etc. According to the model setup described in Statistical Analysis Section, the underlying assumption was that the early-stage activities were the main causes of reproductive traits, which may not be true. I strongly encourage the authors to reconsider the statistical analysis.
Line 345-346: Another limitation might be the measures from NUtrack. The quantities were generated from an automatic tracking and behavior detection system. It was not discussed whether the measures were validated or could be considered as ground truth.
Author Response
The study presented by Millburn et al. aims at examining the association between early-stage activities of gilts and reproductive traits at later life stages. This work is of great interest to swine farming, management, and breeding. The introduction section is appropriate, logical, and sound. The objective of the study is well-defined. However, in my perspective, the statistical analysis and/or regression model setup can be improved, while the results might be updated depending on the revised analysis. My specific comments are below.
a. The statistical analysis section has been reworked to show greater depth and clarity to the analysis performed.
Lines 132-135: the measures and quantifications are collected by deep learning models. I would suggest the authors indicate that the measures are not exactly the same as ground truth measures.
a. Added clarity to the section to note that the system was trained on human annotated images/videos as well as validated. While not exactly ground truth, it is based on ground truth.
Line 161: presenting the linear equations in a formula form is encouraged. For the linear model, did it account for effects in addition to the activity variables e.g., social group and body weight (or weight gain) during the observation period at the early stage? If yes, please list all the effects included in the model. If not, please justify the reason for not adjusting for those variables. Overall, there is a lack of information to justify the statistical model.
a. The statistical analysis section has been reworked to include the formula form of the equation and all items that were included in the original analysis as well as why they were not included in the final model.
Line 187-189: For the highly correlated variables e.g., angle and distance (referred to correlations shown in Table 3), what is the reason for including both variables in the regression model? Please justify.
a. We added a note to clarify our intentions behind reporting the simple linear regression analysis for all the activity traits given the novelty of the traits analyzed. Had multiple regression models shown an improvement in accuracy, we would agree with the reviewer that adding multiple highly correlated traits to such a model would not result in sound reportable data.
Line 244-245: the results based on the statistical analysis described in this manuscript may be inaccurate. There was a time difference between the measures of the early-stage activities and the measures of the reproductive traits. It would be risky to omit effects such as animal ID, social group, weight (or weight gain), etc. According to the model setup described in Statistical Analysis Section, the underlying assumption was that the early-stage activities were the main causes of reproductive traits, which may not be true. I strongly encourage the authors to reconsider the statistical analysis.
a. The reworked section on the statistical analysis section should alleviate much of this concern regarding what is included in the analysis. The initial sentence to the discussion section was also reworded to indicate we are not suggesting these activities are the only indicator of reproductive success but are noting these activities do add to the scientific knowledge of what could be indicators for reproductive success.
Line 345-346: Another limitation might be the measures from NUtrack. The quantities were generated from an automatic tracking and behavior detection system. It was not discussed whether the measures were validated or could be considered as ground truth.
a. We included statements to acknowledge that while the system was trained and annotated on human annotated images and videos as well as validated by humans, that each second was not watched to ensure the system was 100% accurate/ground truth each second. We understand the hesitancy of those not familiar with the system to question the accuracy of the system though the system was validated in previous manuscripts. We would like to point out that there is ~290,304,000 seconds of data utilized for this manuscript that is not feasible to be ground truth checked in its entirety via human annotation.
Round 2
Reviewer 2 Report
I appreciate the authors significant revisions addressing considerations to describing hypothesis of tracking finishing pigs to pig behavior and the justification translation of human derived algorithms to pigs as well as update literature citations. I believe this publication adds to the knowledge of animal fitness to reproductive performance. I appreciate the opportunity to review your paper and its revisions.
Author Response
We thank the reviewer for their diligence and previous suggestions. We feel our manuscript is better for it.
Reviewer 3 Report
The current form of the manuscript has been improved significantly in comparison to the previous one. The authors addressed most of my review comments except for the comments related to statistical analysis. I would recommend the paper for publication once the statistical analysis and results are presented in a more rigorous form. Please see the comments below:
1. I was a bit confused after reading the revised section (Statistical Analysis). The traits mentioned in Line 194 were dropped from the regression model. It would make sense if none of them had significant effects (larger p-values) and dropped from the model. By reading the text, it implied that there might be some traits that were significant but claimed to be confounding variables, which were dropped from the model anyway. The authors are recommended to make this part crystal clear. For example, they can show the p-values of those traits and justify why some of them were considered confounding effects. Otherwise, I may assume that by including the traits (mentioned in Line 194) the significance of the NUtrack traits will alter and more likely, turn out to be non-significant.
2. In Line 205 X was claimed to be one of the NUtrack traits, which implies that each Y (GL, NBA, or NW) was regressed on one measurement and there were 3x7=21 models in total. However, in Line 226 it looked like all NUtrack measures were fitted in the regression model (3 models in total). Then my question is, which model was used to obtain the results in Table 4?
3. Line 209: how were the models compared and which models were compared? Why was anova() used for model comparison instead of ANOVA?
4. Line 230: when there are correlated variables, the variables can be processed depending on the model and its assumption. The authors utilized simple linear regression models, which assume the predictor(s) X to be i.i.d variables. Including correlated/confounding variables will negatively affect the interpretability and the predictive performance of the model. However, the authors indicated the goal would be to gain knowledge of the activity traits. The statement underlay that the correlated variables should remain in the model anyway. If this is the case, why did the authors use simple linear regression models without specifying variance-covariance structures in the statistical model?
Author Response
We thank the reviewer for the rigorous review of the manuscript. We utilized the reviewer's comments and suggestions to further enhance and clarify the manuscript.
- I was a bit confused after reading the revised section (Statistical Analysis). The traits mentioned in Line 194 were dropped from the regression model. It would make sense if none of them had significant effects (larger p-values) and dropped from the model. By reading the text, it implied that there might be some traits that were significant but claimed to be confounding variables, which were dropped from the model anyway. The authors are recommended to make this part crystal clear. For example, they can show the p-values of those traits and justify why some of them were considered confounding effects. Otherwise, I may assume that by including the traits (mentioned in Line 194) the significance of the NUtrack traits will alter and more likely, turn out to be non-significant.
- Section was rewritten to specifically note the p values of the traits dropped from further analysis were are > 0.1. The comment on confounding was removed as the traits were not significant and therefore there was not a need to note any confounding effects.
- In Line 205 X was claimed to be one of the NUtrack traits, which implies that each Y (GL, NBA, or NW) was regressed on one measurement and there were 3x7=21 models in total. However, in Line 226 it looked like all NUtrack measures were fitted in the regression model (3 models in total). Then my question is, which model was used to obtain the results in Table 4?
- Regression analysis was originally only analyzed on total lie time and not split between lie sternal and lie lateral, thus 7 NUtrack traits. After further analysis and the correlation was noted between sternal and lateral, we ran sternal and lateral independently and failed to make the correction from 7 traits to 8 traits that creates this error/confusion. Changed to note there are 8 NUtrack traits. Therefore there are 24 total models (8 NUtrack traits and 3 reproductive traits) as table 4 indicates as each reproductive trait and NUtrack were ran independently. This is further clarified in the results section (starting at line 225).
- Line 209: how were the models compared and which models were compared? Why was anova() used for model comparison instead of ANOVA?
- anova() is the notation of the script for ANOVA using the R stats program.
- Line 230: when there are correlated variables, the variables can be processed depending on the model and its assumption. The authors utilized simple linear regression models, which assume the predictor(s) X to be i.i.d variables. Including correlated/confounding variables will negatively affect the interpretability and the predictive performance of the model. However, the authors indicated the goal would be to gain knowledge of the activity traits. The statement underlay that the correlated variables should remain in the model anyway. If this is the case, why did the authors use simple linear regression models without specifying variance-covariance structures in the statistical model?
- Results section 225-237 was reworded to ensure the true analysis performed came across clearer as this results section focused on independent simple linear regression and not correlated traits. Any mention of correlation was removed from this section.